# Immunotherapy in GI Cancers: Lessons from Key Trials and Future Clinical Applications

**DOI:** 10.3390/antib14030058

**Published:** 2025-07-11

**Authors:** Supriya Peshin, Faizan Bashir, Naga Anvesh Kodali, Adit Dharia, Sajida Zaiter, Sakshi Singal, Nagaishwarya Moka

**Affiliations:** 1Norton Community Hospital, Ballad Health, Norton, VA 24273, USA; 2School of Medicine, Shiraz University of Medical Sciences, Shiraz 71345-3119, Iran; faizanbashir.med@gmail.com (F.B.); susuzaiter22332@gmail.com (S.Z.); 3HCA Florida Ocala Hospital, University of Central Florida, Ocala, FL 34471, USA; anveshkodali@gmail.com; 4HCA Florida West Hospital, Pensacola, FL 32514, USA; aditdha@gmail.com; 5Medical Oncology, East Tennessee State University, Johnson City, TN 37604, USA; singalsakshi1@gmail.com; 6Debusk College of Osteopathic Medicine, Lincoln Memorial University, Harrogate, TN 37752, USA

**Keywords:** immunotherapy, gastrointestinal cancers, immune checkpoint inhibitors (ICIs), PD-1/PD-L1 pathway, microsatellite instability-high (MSI-H) deficient mismatch repair (dMMR)

## Abstract

Immunotherapy has emerged as a transformative approach in gastrointestinal (GI) cancers, addressing historically poor survival rates in advanced-stage disease. Immune checkpoint inhibitors (ICIs) targeting the PD-1/PD-L1 axis demonstrate remarkable efficacy in colorectal cancer with deficient mismatch repair (dMMR) or high microsatellite instability (MSI-H), exemplified by trials like NICHE-2 achieving exceptional pathological response rates. However, significant limitations persist, including resistance in some dMMR/MSI-H tumors, minimal efficacy in proficient mismatch repair (pMMR) tumors, and low overall response rates across most GI malignancies due to tumor heterogeneity and immune evasion mechanisms. Predictive biomarkers such as tumor mutational burden (TMB) and PD-L1 expression are crucial for optimizing patient selection, while hypermutated pMMR tumors with POLE mutations represent emerging therapeutic opportunities. In pancreatic adenocarcinoma, where survival remains dismal, combination strategies with chemotherapy and novel approaches like cancer vaccines show promise but lack transformative breakthroughs. Esophagogastric cancers benefit from ICIs combined with chemotherapy, particularly in MSI-H and HER2-positive tumors, while hepatocellular carcinoma has achieved significant progress with combinations like atezolizumab–bevacizumab and durvalumab–tremelimumab surpassing traditional therapies. Biliary tract cancers show modest improvements with durvalumab–chemotherapy combinations. Despite these advances, immunotherapy faces substantial challenges including immune-related adverse events, acquired resistance through cancer immunoediting, and the need for biomarker-driven approaches to overcome tumor microenvironment barriers. This review discusses key clinical trials, therapeutic progress, and emerging modalities including CAR T-cell therapies and combination strategies, emphasizing the critical need to address resistance mechanisms and refine precision medicine approaches to fully realize immunotherapy’s potential in GI malignancies.

## 1. Introduction

Gastrointestinal (GI) cancers, including gastric, esophageal, colorectal, pancreatic, and liver cancers, are among the most common and lethal cancers worldwide. According to GLOBOCAN 2020, colorectal cancer ranks as the third most frequently diagnosed cancer globally and the second leading cause of cancer-related death, while gastric and liver cancers account for a major share of cancer mortality, particularly in East Asia and parts of South America [1]. In the United States, colorectal cancer (CRC) is the third most common cancer in both men and women. Pancreatic and liver cancers, though less common, exhibit among the lowest five-year survival rates [2].

In recent years, immunotherapy has become an important treatment option for several GI cancers. Immune checkpoint inhibitors (ICIs), particularly those targeting programmed death-1 (PD-1), programmed death-ligand 1 (PD-L1), and cytotoxic T-lymphocyte-associated protein 4 (CTLA-4), have demonstrated remarkable clinical benefits across multiple malignancies by enhancing the immune system’s ability to recognize and attack tumor cells [3,4]. Numerous clinical trials support the efficacy of ICIs in GI cancers, particularly in patients whose tumors display microsatellite instability-high (MSI-H) or high PD-L1 expression. For example, the KEYNOTE-059 trial demonstrated the efficacy of pembrolizumab in advanced gastric cancer [5], the CheckMate-649 trial showed the benefit of nivolumab plus chemotherapy in gastroesophageal adenocarcinoma [6], and the KEYNOTE-177 trial established pembrolizumab as a first-line treatment in MSI-H colorectal cancer [7].

Immunotherapy works by reactivating immune responses that tumors suppress. Under physiologic conditions, immune checkpoints such as PD-1 and CTLA-4 regulate immune responses to prevent excessive inflammation and autoimmunity [8]. Tumor cells exploit these pathways to evade immune detection. PD-1, a receptor on T-cells, interacts with PD-L1 or PD-L2 on tumor or immune cells to dampen T-cell activity. CTLA-4, another checkpoint receptor, blocks T-cell activation at an earlier stage in the lymph nodes [3,8]. As described by Peshin et al., pembrolizumab, a PD-1 inhibitor, binds to the PD-1 receptor on T-cells, blocking its interaction with PD-L1 and PD-L2, thereby reactivating T-cells to target tumors [9]. Unlike cytotoxic agents, pembrolizumab does not directly kill tumor cells but enhances the immune system’s capacity to do so. Figure 1 depicts how pembrolizumab alleviates T-cell inhibition, enabling immune-mediated tumor destruction.

Understanding these mechanisms helps explain both the successes and limitations of immunotherapy. It also supports ongoing research to improve outcomes, such as combining ICIs with other treatments and developing biomarkers to identify patients most likely to benefit [10].

## 2. Immunotherapy in Gastric and Gastroesophageal Junction Cancers

The treatment landscape of gastric and gastroesophageal junction (GEJ) cancers has evolved considerably with the advent of immunotherapy, particularly with the use of ICIs targeting PD-1/PD-L1. Several clinical trials have demonstrated the safety and efficacy of these agents, both as monotherapies and in combination with chemotherapy.

Pembrolizumab, a monoclonal antibody targeting PD-1, has emerged as a promising therapeutic option for patients with advanced gastric and GEJ cancers. In the Phase 1b KEYNOTE-012 trial, pembrolizumab was evaluated in patients with PD-L1-positive advanced or metastatic gastric cancer who had previously undergone standard chemotherapy [11]. Pembrolizumab was administered intravenously every two weeks until disease progression or unacceptable toxicity. The study demonstrated that pembrolizumab elicited clinically meaningful responses, particularly in PD-L1-positive tumors. The overall response rate (ORR) was significantly higher in these patients, supporting the role of PD-L1 expression as a predictive biomarker. The treatment showed manageable toxicity and notable anti-tumor activity [11].

Building on these findings, the Phase 2 KEYNOTE-059 trial evaluated pembrolizumab in chemotherapy-refractory advanced gastric cancer across three cohorts [5,12]. In Cohort 1, patients received pembrolizumab monotherapy (200 mg every three weeks) after at least two prior chemotherapy regimens. In Cohort 2, pembrolizumab was combined with 5-fluorouracil (5-FU) and cisplatin in patients with PD-L1-positive, HER2-negative advanced gastric cancer. Cohort 3 included treatment-naïve patients who received pembrolizumab monotherapy. The ORR across all patients was approximately 11.6%, with durable responses noted in a subset. Among PD-L1-positive tumors, response rates were notably higher, further supporting PD-L1 as a predictive biomarker. Progression-free survival (PFS) was improved, especially in patients with PD-L1-positive tumors. Notably, pembrolizumab plus chemotherapy showed promise as a first-line therapy across PD-L1 expression levels. Pembrolizumab monotherapy also showed encouraging efficacy in patients with PD-L1 Combined Positive Score (CPS) ≥ 1 [12].

Nivolumab, another PD-1 inhibitor, has shown significant activity in advanced gastric cancer, as demonstrated in the ATTRACTION-2 Phase 3 trial [13]. This study evaluated nivolumab versus placebo in patients with advanced gastric cancer who had received at least two prior lines of chemotherapy. Patients were administered nivolumab (3 mg/kg) or placebo intravenously every two weeks in six-week cycles. Nivolumab treatment led to a significant improvement in overall survival (OS), making it the first Phase 3 trial to show survival benefit in this setting. Although the improvement in PFS was modest initially, it became more apparent approximately two months into treatment, favoring the nivolumab arm [13].

The ATTRACTION-4 trial assessed the benefit of nivolumab in combination with oxaliplatin-based chemotherapy in patients with previously untreated, HER2-negative, unresectable, or recurrent gastric cancer [14]. Patients received oxaliplatin combined with either S-1 or capecitabine, along with nivolumab or placebo. The addition of nivolumab improved PFS, though OS benefits did not reach statistical significance. However, a three-year follow-up showed sustained PFS, duration of response (DOR), and favorable OS hazard ratios among patients treated with nivolumab. Patients who achieved complete or partial responses had notably high three-year survival rates [14]. The regimen was well-tolerated, with no delayed treatment-related adverse events.

The CheckMate-649 Phase 3 trial further cemented the role of immunotherapy in the frontline setting by comparing nivolumab plus chemotherapy to chemotherapy alone in previously untreated patients with advanced gastric, GEJ, or esophageal adenocarcinoma [6,15]. Patients received nivolumab with either XELOX (capecitabine + oxaliplatin) or FOLFOX (leucovorin + oxaliplatin) regimens or chemotherapy alone. In select arms, nivolumab was also combined with ipilimumab. The combination of nivolumab and chemotherapy demonstrated significant improvements in OS and PFS compared to chemotherapy alone, particularly in patients with PD-L1 CPS ≥ 5 [6,15]. These results supported the use of nivolumab as a new standard-of-care treatment option for advanced gastric and GEJ adenocarcinomas, with an acceptable safety profile.

The CheckMate-032 Phase 2 trial explored the use of nivolumab alone and in combination with ipilimumab in patients with chemotherapy-refractory advanced gastric and GEJ cancers [16]. Patients were randomized to receive nivolumab monotherapy or one of two nivolumab–ipilimumab combination regimens. Both monotherapy and combination therapy showed durable anti-tumor responses. The combination of nivolumab 1 mg/kg plus ipilimumab 3 mg/kg demonstrated the highest ORR (24%), although median OS was similar across arms [16]. These results indicate that nivolumab, alone or in combination with ipilimumab, may offer meaningful therapeutic benefit in heavily pretreated esophagogastric cancer.

Multiple meta-analyses have further synthesized evidence from clinical trials to compare different immunotherapy strategies. A comprehensive network meta-analysis led by Zhang et al., which included 7898 patients across eight trials, identified cadonilimab plus chemotherapy as a leading treatment option for HER2-negative advanced gastric cancer, especially in PD-L1 CPS-positive patients [17]. Another meta-analysis by Pan et al., incorporating data from five studies with 1730 patients, recommended nivolumab 3 mg/kg every two weeks or the combination of nivolumab 1 mg/kg plus ipilimumab 3 mg/kg every three weeks as the preferred regimens based on their high efficacy and favorable safety profiles [18].

Qian et al. evaluated 13 studies involving 2841 patients to assess the impact of PD-1/PD-L1 inhibitors combined with neoadjuvant chemotherapy (NCT) in resectable, locally advanced gastric cancer, AEG, and esophageal cancer [19]. While immunochemotherapy improved pathological responses, its effect on long-term survival remained limited, warranting further investigation. Similarly, a Bayesian network meta-analysis by Liu et al., including 31 trials, confirmed the benefit of combining PD-1 inhibitors with chemotherapy as a first-line strategy for advanced gastric cancer. The analysis highlighted this approach as effective even in patients lacking HER2 or CLDN18.2 expression, with or without PD-L1 positivity [20].

Collectively, these trials and analyses have transformed the therapeutic landscape of gastric and GEJ cancers. The integration of PD-1 inhibitors, both as monotherapy and in combination with chemotherapy or CTLA-4 inhibitors, has provided durable clinical benefits and is progressively establishing new standards of care in various disease settings. A summary of key clinical trials evaluating immune checkpoint inhibitors in gastric and gastroesophageal junction cancers is presented in Table 1.

Comparative studies such as KEYNOTE-059 and CheckMate-649 highlight meaningful response differences driven by PD-L1 expression, yet the variability in CPS thresholds (≥1 vs. ≥5) creates inconsistencies in patient selection. Moreover, the benefit of ICI therapy appears greater in HER2-negative and MSI-H subsets, while trials like ATTRACTION-4 failed to show OS benefit despite PFS improvement, likely due to regional patient differences (predominantly Asian cohort). These disparities, along with modest efficacy in PD-L1-negative or CLDN18.2-unselected populations, underscore the need for better stratification tools and unified biomarker criteria.

## 3. Immunotherapy in Hepatocellular Carcinoma (HCC)

Hepatocellular carcinoma (HCC) ranks as the third leading cause of cancer-related death worldwide, with many patients diagnosed at advanced stages [21]. Effective HCC management requires careful consideration of tumor extent, comorbidities, and liver function [22]. Early-stage HCC can often be treated with local ablation, surgical resection, or liver transplantation [22]. However, residual tumor cells often persist after ablation [23,24], and the utility of surgical resection is limited in patients with advanced-stage disease [25]. Furthermore, targeted therapies are frequently insufficient, often leading to the development of drug resistance [26]. ICIs have demonstrated efficacy in patients with advanced-stage HCC [27].

A pilot study published in 2013 evaluated tremelimumab (anti-CTLA-4) in patients with advanced HCC and chronic hepatitis C infection [28]. Tremelimumab showed a partial response rate of 17.6% and disease control rate of 76.4%, with a median time to progression of 6.48 months. Notably, it also reduced HCV viral load and enhanced anti-HCV immune responses, with an acceptable safety profile in cirrhotic patients.

The CheckMate 040 trial, a multicenter Phase I/II study published in 2017, evaluated nivolumab, an anti-PD-1 monoclonal antibody, for the treatment of advanced HCC and showed promising results [29]. Based on these findings, the FDA approved nivolumab in September 2017 as a second-line treatment for advanced HCC in patients previously treated with sorafenib. Subsequently, the Phase III CheckMate 459 trial, completed in June 2019, compared nivolumab with sorafenib. While the initial results did not show a statistically significant survival benefit, they suggested that nivolumab could be a viable option for patients in whom tyrosine kinase inhibitors (TKIs) and antiangiogenic drugs are contraindicated or pose substantial risks [30]. Notably, the long-term follow-up from CheckMate 459 reported in 2020 demonstrated a clinically meaningful survival benefit with nivolumab [31]. Similarly, the KEYNOTE-224 trial led to the FDA approval of pembrolizumab in 2018 for patients with advanced HCC previously treated with sorafenib [32]. The KEYNOTE-937 trial is an ongoing Phase III study evaluating pembrolizumab versus placebo as adjuvant therapy in patients with hepatocellular carcinoma (HCC) who have achieved a complete radiological response after surgical resection or local ablation ([1]) [33]. The trial aims to assess the safety and efficacy of pembrolizumab in reducing recurrence risk and improving survival outcomes in this high-risk population.

In 2020, a Phase 1b trial (the GO30140 study) demonstrated that the combination of atezolizumab and bevacizumab improved overall and progression-free survival compared to sorafenib in patients with unresectable HCC [34]. The Phase III IMbrave150 studies further confirmed these findings, showing that patients with unresectable HCC experienced significantly better overall and progression-free survival with atezolizumab and bevacizumab compared to sorafenib [35]. As a result, this combination of atezolizumab and bevacizumab has become the standard of care for advanced-stage HCC [35].

The IMbrave050 Phase III trial (2022) evaluated adjuvant atezolizumab plus bevacizumab versus active surveillance in 668 patients with high-risk HCC after curative resection or ablation [36]. The combination significantly improved recurrence-free survival (hazard ratio 0.72; *p* = 0.012) at a median follow-up of 17.4 months. This is the first Phase III study to show positive results for adjuvant therapy in HCC, although longer follow-up is needed to assess overall survival outcomes.

The EMERALD-2 Phase III global trial (NCT03847428), initiated in 2019, is evaluating adjuvant durvalumab with or without bevacizumab versus placebo in patients with HCC at high risk of recurrence after curative hepatic resection or ablation [37]. This randomized, double-blind, placebo-controlled study aims to assess efficacy and safety, with recurrence-free survival as the primary endpoint and secondary outcomes including overall survival, time to recurrence, and quality of life.

The HIMALAYA trial, a global Phase III study published in 2023, evaluated the STRIDE regimen, a single high-priming dose of tremelimumab (anti-CTLA-4) plus durvalumab (anti–PD-L1) in patients with unresectable HCC and no prior systemic therapy [38]. The trial randomized 1171 patients to receive STRIDE, durvalumab monotherapy, or sorafenib. The STRIDE regimen significantly improved median OS to 16.43 months compared to 13.77 months with sorafenib (hazard ratio [HR] 0.78, *p* = 0.0035). Durvalumab monotherapy was noninferior to sorafenib (median OS 16.56 months, HR 0.86), providing a viable alternative for patients unsuitable for combination strategies. Grade 3/4 adverse events occurred in 50.5% (STRIDE), 37.1% (durvalumab), and 52.4% (sorafenib) of patients. These findings establish STRIDE as an effective first-line immunotherapy regimen for unresectable HCC.

The combination of ICIs with TKIs has also been explored. A Phase Ib study investigated the use of durvalumab in combination with ramucirumab, an anti-VEGFR-2 IgG1 monoclonal antibody [39]. This study showed that this combination was safe and showed anti-tumor activity in patients with HCC. Finally, a multicenter, open-label Phase I/II study is currently evaluating the safety and efficacy of MGD013—an anti-PD-1/anti-LAG-3 Dual-Affinity Re-Targeting (DART) protein—both as monotherapy and in combination with brivanib, a selective dual inhibitor of VEGFR and fibroblast growth factor receptors (FGFR), in patients with advanced liver cancer (NCT04212221) [40]. All in all, in recent years, ICIs have been shown to be an effective treatment option for patients with advanced-stage HCC. Table 2 provides an overview of landmark clinical trials evaluating immunotherapy in HCC.

While the IMbrave150 trial established atezolizumab–bevacizumab as a first-line standard for unresectable HCC, CheckMate-459 did not meet statistical significance for OS despite a clinically meaningful trend favoring nivolumab. These differing outcomes may reflect variations in patient liver function, viral hepatitis etiology, or trial design. Biomarker development in HCC remains rudimentary, with no established predictive markers for ICI response. In contrast, recent efforts in intrahepatic cholangiocarcinoma (iCCA) have emphasized the importance of optimizing tissue procurement for molecular testing and integrating ctDNA analyses, within a multidisciplinary, precision-medicine framework to guide targeted therapies [42]. Additionally, real-world barriers such as bleeding risk from varices complicate bevacizumab use, limiting the universal applicability of successful regimens.

## 4. Immunotherapy in Colorectal Cancer (CRC)

CRC remains one of the leading causes of cancer-related mortality worldwide. Immunotherapy, particularly ICIs, has significantly transformed the treatment landscape for many malignancies. However, its efficacy in CRC is highly dependent on the molecular characteristics of the tumor. The two primary immunologic subtypes of CRC are microsatellite instability-high (MSI-H) and microsatellite stable (MSS), which exhibit markedly different responses to ICIs.

Microsatellite instability-high (MSI-H) CRC results from defects in DNA mismatch repair (dMMR) genes, leading to a high tumor mutational burden (TMB) and the formation of numerous neoantigens. This enhanced immunogenicity makes MSI-H tumors particularly responsive to PD-1 blockade, with checkpoint inhibitors such as pembrolizumab and nivolumab demonstrating remarkable efficacy [43].

Conversely, microsatellite-stable (MSS) CRC, which accounts for approximately 85% of all CRC cases, has an intact mismatch repair system and a low mutational burden. These tumors are often described as ‘immune cold’ characterized by low T-cell infiltration and an immunosuppressive tumor microenvironment (TME), rendering them largely resistant to ICIs [44]. As such, addressing immunotherapy resistance in MSS CRC remains a major challenge, with ongoing research into innovative strategies to enhance immune activation, including TGF-β blockade, microbiome modulation, and combination therapies.

Checkpoint inhibitors have revolutionized the management of MSI-H CRC. For instance, pembrolizumab demonstrated superior PFS in the KEYNOTE-177 trial, establishing PD-1 inhibition as a first-line treatment for MSI-H metastatic CRC. Despite this breakthrough, around 30% of patients showed primary or acquired resistance, underscoring the need for combination approaches [7].

Similarly, the CheckMate-142 trial highlighted the efficacy of combining nivolumab (a PD-1 inhibitor) with ipilimumab (a CTLA-4 inhibitor), showing a 31% complete response rate. This combination strategy broadens immune response and enhances T-cell activation, providing an effective option for patients with high-risk features or a resistance to PD-1 monotherapy [45].

The BREAKWATER trial further evaluated the combination of encorafenib (a BRAF inhibitor) and cetuximab (an anti-EGFR antibody) with or without chemotherapy in patients with BRAF V600E-mutant metastatic CRC. This combination demonstrated a significant improvement in the objective response rate (ORR), leading to the FDA’s accelerated approval for its use, offering a targeted therapeutic option [46].

The NICHE-2 trial demonstrated the transformative potential of neoadjuvant immunotherapy in locally advanced deficient mismatch repair (dMMR) colon cancer. This Phase II study treated 115 patients with one dose of ipilimumab (1 mg/kg) and two doses of nivolumab (3 mg/kg) ≤ 6 weeks prior to surgery, achieving a 100% 3-year disease-free survival and high pathological response rates. The clinical significance of these results becomes evident when compared to conventional therapy, as pathologic response using standard of care FOLFOX is approximately 5% in dMMR. This dramatic improvement establishes neoadjuvant checkpoint inhibition as a new standard of care for locally advanced dMMR colon cancer, representing a paradigm shift from chemotherapy-based approaches to immunotherapy-first strategies in this molecularly defined subset of patients [47,48,49].

In addition, the FRESCO and FRESCO-2 trials assessed the efficacy of fruquintinib, a selective VEGFR inhibitor, in refractory metastatic CRC. These trials established fruquintinib as a new treatment option for patients who had exhausted standard therapies, addressing an unmet need in advanced CRC care [50]. A summary of important clinical trials shaping immunotherapy strategies in colorectal cancer is shown in Table 3.

Despite the remarkable progress in MSI-H CRC, resistance to immunotherapy persists in many cases [54]. To overcome this challenge, innovative combination strategies are being explored, included in Figure 2 [54]:

In MSI-H/dMMR CRC, immune checkpoint blockade has proven transformative, with pembrolizumab in KEYNOTE-177 and nivolumab ± ipilimumab in CheckMate-142 demonstrating high response rates and durability. However, up to 30% of patients exhibit primary or acquired resistance, emphasizing disease heterogeneity even within this biomarker-defined group. MSS CRC, which constitutes the majority of CRC cases, remains refractory to ICIs, reflecting its immunologically “cold” nature. Biomarker limitations—such as variable TMB thresholds and a lack of reliable indicators in MSS disease—further restrict the reach of immunotherapy in CRC.

### 4.1. MSS CRC: Overcoming Immunotherapy Resistance

MSS CRC presents a substantial challenge in immunotherapy due to its low mutational burden and immune-excluded phenotype [55]. The TME of MSS CRC fosters immune evasion through mechanisms such as TGF-β signaling, regulatory T-cells (Tregs), and myeloid-derived suppressor cells (MDSCs). As a result, single-agent ICIs have shown little to no clinical benefit [55]. Innovative strategies being studied are included in Figure 3:

### 4.2. Clinical Decision-Making: When to Escalate Therapy in MSI-H CRC

Optimal clinical decision-making in MSI-H CRC requires careful consideration of resistance mechanisms and timely escalation of therapy. The primary goal is to achieve durable responses while addressing primary or acquired resistance effectively. The recommended approaches include:

*First-Line Therapy:* Pembrolizumab, as established by the KEYNOTE-177 trial, or nivolumab with or without ipilimumab, as demonstrated in the CheckMate-142 trial, represent the standard initial choices. These therapies offer significant progression-free survival, yet some patients may not respond initially or may develop resistance [7,45].

*Escalation Strategy for Primary Resistance:* When primary resistance occurs, a dual checkpoint blockade using CTLA-4 and PD-1 inhibitors may enhance T-cell activation and broaden the immune response [56]. Alternatively, VEGF inhibitors like bevacizumab or novel LAG-3 inhibitors may help overcome resistance by modulating the tumor microenvironment [57,58].

*Management of Acquired Resistance:* In cases where resistance emerges after initial response, therapeutic strategies may include DNA damage response (DDR) inhibitors such as PARP inhibitors, or the use of tumor-infiltrating lymphocyte (TIL)-based approaches to rejuvenate immune engagement [59,60].

### 4.3. Microbiome Modulation and Vitamin D: A Gut Feeling About Immunotherapy?

The gut microbiome plays an increasingly recognized role in modulating the effectiveness of immunotherapy in CRC. Certain bacterial species, such as *Akkermansia muciniphila* and *Bacteroides fragilis*, have been associated with enhanced responses to ICIs [61]. Emerging strategies aim to manipulate the microbiome to improve therapeutic outcomes, including fecal microbiota transplantation (FMT) from responders to non-responders [61].

Vitamin D, known for its role in bone health and immunity, is gaining attention for its potential to influence outcomes in CRC. Vitamin D deficiency has been correlated with a worse CRC prognosis, and supplementation may enhance T-cell activation and improve immune responses [62,63]. Immunotherapy has undeniably transformed CRC management, but challenges remain, particularly in MSS CRC. Combining an immune checkpoint blockade with targeted therapies, leveraging microbiome modulation, and enhancing TME normalization are promising avenues to improve outcomes. Exploring predictive biomarkers, incorporating machine learning for treatment optimization, and developing combinatorial approaches will further strengthen the potential of immunotherapy in CRC. The continued exploration of novel strategies and personalized approaches holds the potential to revolutionize immunotherapy in CRC, ultimately improving patient survival and quality of life.

## 5. Immunotherapy in Pancreatic Cancer

Pancreatic ductal adenocarcinoma (PDAC) remains among the most formidable malignancies in oncology, characterized by an aggressive clinical course, late-stage presentation, and notable resistance to conventional therapeutic modalities. Despite the transformative success of immunotherapy in various solid tumors, its impact in PDAC has been modest. This limited efficacy stems primarily from the tumor’s profoundly immunosuppressive microenvironment, inherently low tumor mutational burden (TMB), and poor immune cell infiltration. In light of these challenges, ongoing research is evaluating combination strategies such as ICIs with stroma-modulating agents, chimeric antigen receptor (CAR) T-cell therapy, and KRAS-targeted approaches.

### 5.1. Barriers to Immunotherapy in PDAC

#### 5.1.1. Immunosuppressive Tumor Microenvironment

PDAC is distinguished by an intensely fibrotic and immunosuppressive tumor microenvironment, which acts as a major barrier to effective immune-mediated therapies. The desmoplastic stroma—comprising extracellular matrix proteins, cancer-associated fibroblasts (CAFs), endothelial cells, and various immune populations—often outweighs the neoplastic epithelial component [64,65]. This stromal density impairs T-cell infiltration [66] and contributes to immune evasion. Moreover, regulatory T-cells (Tregs), myeloid-derived suppressor cells (MDSCs), and immunosuppressive cytokines, such as transforming growth factor-beta (TGF-β) and interleukin-10 (IL-10), predominate within the tumor, further hampering anti-tumor immunity.

#### 5.1.2. Low Tumor Mutational Burden

Compared to other malignancies, PDAC exhibits a relatively low TMB, which limits the generation of neoantigens and consequently reduces immunogenicity and responsiveness to immune checkpoint blockade (ICB) [67,68].

#### 5.1.3. Poor Immune Cell Infiltration

PDAC is largely immunologically “cold”, typified by limited T-cell infiltration and low PD-L1 expression [69,70]. Additionally, the tumor supports an immune milieu dominated by suppressive cell types—Tregs, dendritic cells, MDSCs, and M2-polarized macrophages—all of which contribute to a poor prognosis [71,72]. These features, alongside a low neoantigen load and an inflammatory, hypoxic microenvironment, synergistically diminish the efficacy of monotherapy immunotherapeutic agents [73,74].

### 5.2. Immunotherapeutic Strategies in Pancreatic Cancer

#### 5.2.1. ICIs in Combination with Stroma-Modulating Agents

To counteract PDAC’s dense stromal architecture, several combination strategies are under investigation to augment ICI efficacy. These include

Targeting key oncogenic drivers such as KRAS, NRG1, and NTRK [75,76];Restoring tumor suppressor activity involving TP53, CDKN2A, and SMAD4 [77];Modulating the TGF-β–SMAD4 signaling axis to influence stromal and immune interactions [77].

Among these, the CXCL12/CXCR4 signaling pathway has emerged as a crucial regulator of immune evasion in PDAC. CXCR4 blockade enhances T-cell infiltration and improves sensitivity to PD-1/PD-L1 inhibition [78,79]. Clinical evaluation of CXCR4 antagonists, such as plerixafor (AMD3100), has shown promising results in sensitizing PDAC cells to immunotherapy and chemotherapy, thereby enhancing immune recognition [80].

Similarly, TGF-β inhibitors, including SB525334 [81] and galunisertib [82,83], have demonstrated potential in remodeling the tumor microenvironment and augmenting ICI responses. However, systemic toxicities remain a significant concern that warrants further investigation.

#### 5.2.2. CAR T-Cell Therapy

Adoptive T-cell therapy, particularly CAR T-cell strategies, has gained traction in PDAC. Despite formidable barriers, several tumor-associated antigens—such as mesothelin (MSLN), prostate stem cell antigen (PSCA), carcinoembryonic antigen (CEA), HER2, MUC1, and CD133—are being explored as targets [84]. Mesothelin, expressed in approximately 80–85% of PDAC cases, has emerged as a promising candidate, with several CAR T-cell constructs currently in clinical trials [85]. Innovations in CAR T-cell design—such as second-generation constructs with co-stimulatory domains and dual-targeting capabilities—aim to overcome the immunosuppressive microenvironment [86]. Combination approaches integrating CAR T-cells with ICIs or cytokine modulation are being pursued to further enhance therapeutic efficacy [86].

#### 5.2.3. KRAS-Targeted Therapies

Given that KRAS mutations are nearly ubiquitous in PDAC, targeting this oncogene is a rational therapeutic strategy. While agents like erlotinib combined with gemcitabine have shown only a modest clinical benefit [87,88], newer selective KRAS inhibitors offer renewed promise. Specifically, KRAS-G12C inhibitors such as sotorasib and adagrasib have demonstrated encouraging clinical activity [88,89].

A Phase II trial evaluating sotorasib in KRAS-G12C mutant PDAC showed partial responses and disease stabilization in a significant subset of patients [89]. Moreover, preclinical models suggest that KRAS^G12C inhibition may promote a pro-inflammatory tumor microenvironment, enhancing susceptibility to ICIs [90]. For non-G12C KRAS mutations, combination therapies targeting the MAPK pathway may be necessary due to its role in cellular proliferation and survival [91]. Table 4 summarizes selected clinical trials evaluating immunotherapeutic strategies in pancreatic cancer.

Immunotherapy has largely underperformed in PDAC, as evidenced by low response rates in trials like KEYNOTE-158. The dense desmoplastic stroma and low TMB create a highly immunosuppressive microenvironment, limiting T-cell infiltration and ICI efficacy. Novel strategies combining ICIs with CXCR4 inhibitors (e.g., plerixafor in COMBAT), TGF-β blockers (e.g., galunisertib), or personalized mRNA vaccines (e.g., autogene cevumeran) show early promise but are not yet practice-changing. Furthermore, predictive biomarkers remain elusive, and KRAS-targeted therapies are only effective in a small subset (e.g., KRAS G12C), reinforcing the challenge of generalizing immunotherapy in PDAC.

## 6. Limitations of Immunotherapy in GI Cancers

The immunotherapy rates of response are still low in the most common gastrointestinal malignancies, and intense research is ongoing into the root molecular mechanisms [92]. One of the main reasons behind such variability is tumor heterogeneity, which has a great impact on differential treatment responses. Tumor heterogeneity is largely caused by defective aspects of the tumor microenvironment (TME), such as high mutational burden, hypoxia, and abnormal vasculature [93]. Hypoxia in the TME has been shown to compromise mismatch repair, enhance genomic instability, and enable sub-clonal populations to arise [93,94,95]. Such heterogeneity is a significant barrier to therapeutic success, generating primary as well as acquired resistance to all treatment types [96].

Acquired resistance specifically emerges through a Darwinian selection event, where therapeutic pressure and TME dynamically shift tumor subpopulations with the frequent relapse of resistant clones [97,98]. During immunotherapy, this process is best illustrated through cancer immunoediting involving elimination, equilibrium, and escape phases [97,98]. Although the immune system succeeds in killing malignant cells initially, remaining variants can progress to equilibrium, where immune pressure can change the immunogenicity of the tumor, ultimately leading to the development of immune-evading subclones through processes such as antigen escape [99]. In addition, cancer cells may inhibit immune responses by cytokine release [100] and the expression of surface inhibitory proteins, leading to T-cell exhaustion and anergy [101].

Resistance to immunotherapy with checkpoint blockers is often attributed to changes in the main signaling pathways, particularly those involving PD-1 and its ligands PD-L1 [101]. Interferon-γ has a twofold function in inducing PD-L1 upregulation that facilitates immune escape [102,103,104] and enhances chemokine generation and pro-apoptotic pathways for immune cell invasion and cell death in tumor cells [103,105,106]. Abnormalities in interferon-γ signaling pathways, particularly the JAK/STAT pathway, are ubiquitous in both primary and secondary resistance to immunotherapy [106]. Moreover, CD73 overexpression has been implicated with a poor prognosis and resistance to checkpoint inhibition, making it a potential biomarker and therapeutic target in cancers including pancreatic adenocarcinoma [107,108]. Overall, these elements emphasize the intricate relationship between tumor heterogeneity, immune evasion, and drug resistance in GI malignancies, and the urgent need for new approaches to bypass these obstacles. Immunotherapy does not selectively target tumor cells in isolation but can induce off-target inflammation across organ systems, presenting as immune-related adverse events (irAEs) [109].

Within the GI tract, irAEs may include colitis, enteritis, and hepatitis, which significantly impact patient morbidity and treatment continuity [110]. In addition to these, immune checkpoint inhibitor treatment-related AEs are of major clinical concern, especially when used as combination regimens [111,112]. Frequent AEs are anemia, fatigue, and dysphagia whereas more serious grade 3 or greater events like neutropenia, hypertension, and lymphopenia are also frequently reported. Other significant toxicities include pyrexia, diarrhea, cutaneous reactions, hematologic abnormalities, thyroid disease, endocrinopathies, and pneumonitis [113]. The frequency and severity of these side effects increase significantly when immunotherapy is used in addition to chemotherapy or biologics [113,114]. Combinations that involve anti-CTLA-4 drugs, for instance, are especially likely to cause hypophysitis because CTLA-4 is expressed in the cells of the pituitary gland, leading to potentially permanent damage to the glands if it is not immediately identified [114]. Moreover, combining immunotherapy with VEGF or VEGFR inhibitors increases the risk of hypertension and severe proteinuria, attributed to their anti-angiogenic effects [115,116].

The complex interplay of resistance mechanisms and adverse events underscores the need for more sophisticated therapeutic approaches in GI malignancies. Developing combination strategies with predictive biomarkers will be essential for improving patient outcomes.

## 7. Conclusions

Immunotherapy has significantly reshaped the therapeutic landscape of gastrointestinal cancers, offering hope in malignancies historically marked by poor outcomes. Immune checkpoint inhibitors have demonstrated remarkable efficacy in tumors with high immunogenicity, such as the MSI-H colorectal and certain hepatocellular carcinomas and have shown promise when combined with chemotherapy in esophagogastric cancers. However, substantial challenges remain, particularly in immune-resistant subtypes like microsatellite-stable colorectal and pancreatic cancers. Emerging strategies such as microbiome modulation, TGF-β and VEGF inhibition, dual checkpoint blockade, and epigenetic therapies are under active investigation to overcome resistance and improve immunotherapy responsiveness. As the field evolves, personalized and biomarker-driven approaches will be essential to optimize patient selection and outcomes. Future research should continue to focus on refining combination regimens, identifying predictive biomarkers, and expanding the applicability of immunotherapy across the diverse spectrum of GI malignancies.

## Figures and Tables

**Figure 1 antibodies-14-00058-f001:**
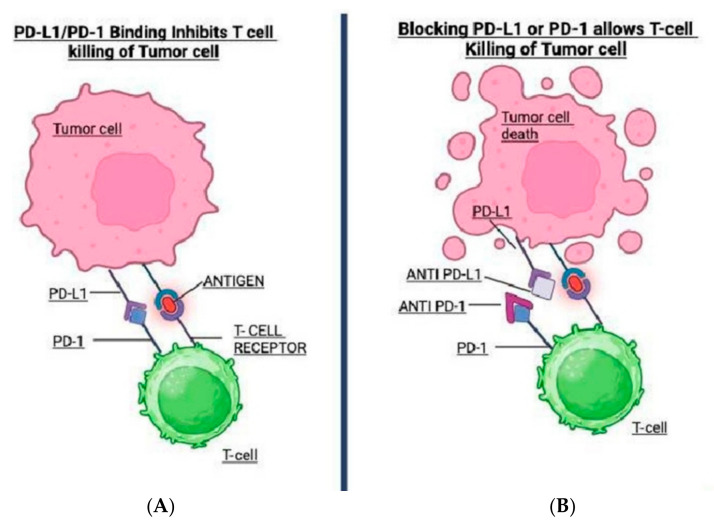
This figure illustrates the mechanism of PD-1/PD-L1 immune checkpoint inhibition in cancer immunotherapy. (**A**): PD-L1 expressed on tumor cells binds to PD-1 receptors on T-cells, inhibiting their cytotoxic activity and allowing tumor immune evasion. (**B**): The use of anti-PD-1 or anti-PD-L1 antibodies blocks this interaction, reactivating T-cell function and enabling effective tumor cell killing. This forms the basis of checkpoint inhibitor therapies like pembrolizumab and atezolizumab.

**Figure 2 antibodies-14-00058-f002:**
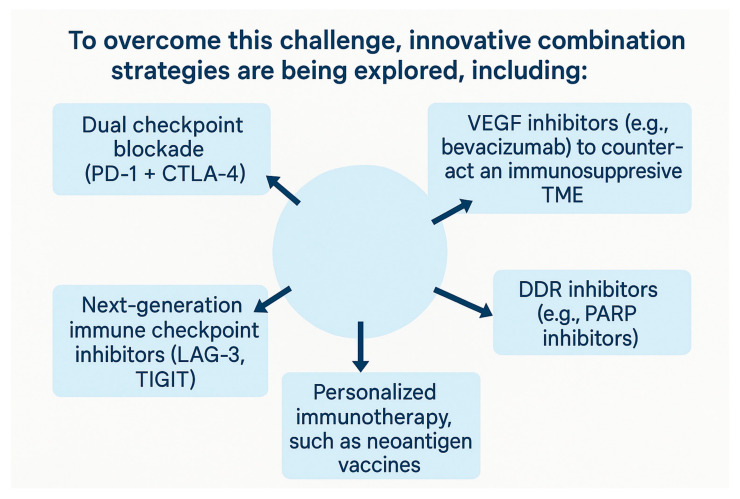
The figure illustrates key innovative combination strategies currently being explored to enhance the efficacy of immunotherapy in cancer treatment. A dual checkpoint blockade, such as the combination of PD-1 and CTLA-4 inhibitors, targets two distinct immune pathways to overcome tumor immune evasion and boost T-cell activation. VEGF inhibitors, like bevacizumab, work by modulating the tumor microenvironment (TME), reducing immunosuppression, and promoting immune cell infiltration into the tumor. Next-generation immune checkpoint inhibitors, including agents targeting LAG-3 and TIGIT, provide additional avenues to reinvigorate exhausted T-cells, especially in cases where traditional PD-1/PD-L1 inhibition is insufficient. DNA damage response (DDR) inhibitors, such as PARP inhibitors, enhance tumor immunogenicity by exploiting defects in DNA repair mechanisms, thereby improving the response to immunotherapy. Lastly, personalized immunotherapies, such as neoantigen vaccines, are designed based on individual tumor-specific mutations to stimulate a precise and robust immune response. Collectively, these strategies aim to overcome resistance mechanisms and expand the benefits of immunotherapy across a broader patient population.

**Figure 3 antibodies-14-00058-f003:**
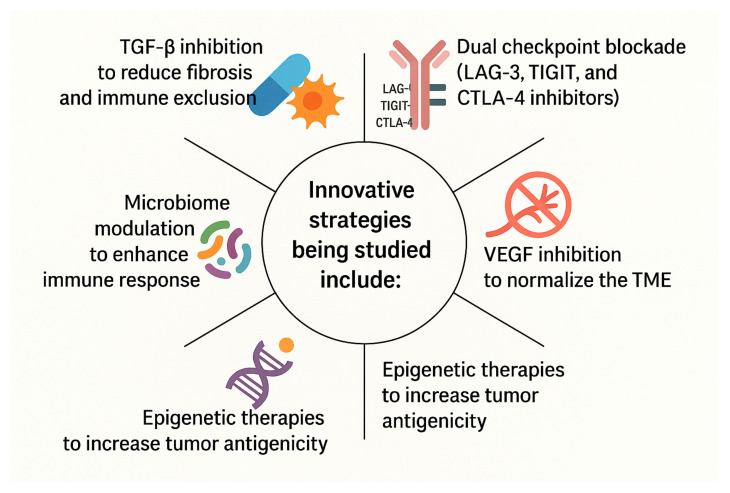
The figure highlights several innovative immunotherapeutic strategies currently under investigation to overcome resistance and improve treatment outcomes. TGF-β inhibition is being explored to reduce fibrosis and alleviate immune exclusion within the tumor microenvironment (TME), thereby enhancing immune cell infiltration. A dual checkpoint blockade, including inhibitors of LAG-3, TIGIT, and CTLA-4, aims to broaden and intensify immune activation by targeting multiple inhibitory pathways simultaneously. VEGF inhibition helps normalize the TME by reducing abnormal vasculature and improving immune accessibility. Modulation of the gut microbiome has emerged as a novel strategy to enhance systemic immune responses and improve immunotherapy efficacy. Lastly, epigenetic therapies are designed to increase tumor antigenicity, making cancer cells more recognizable to the immune system by altering gene expression profiles. Together, these approaches represent a multifaceted effort to enhance the effectiveness of cancer immunotherapy.

**Table 1 antibodies-14-00058-t001:** Gastric and GEJ Cancer Trials.

Trial Name	Phase	Therapy Evaluated	Patient Population/Key Findings
KEYNOTE-012	1b	Pembrolizumab monotherapy	PD-L1+ advanced/metastatic GC after chemotherapy; demonstrated clinical benefit with manageable toxicity.
KEYNOTE-059	2	Pembrolizumab ± chemotherapy	Chemo-refractory or treatment-naïve advanced GC/GEJ cancer; ORR ~11.6%, higher in PD-L1+; durable responses in subsets.
ATTRACTION-2	3	Nivolumab vs. placebo	Advanced GC post ≥ 2 chemotherapy lines; significant OS improvement; first ICI showing survival benefit in this setting.
ATTRACTION-4	2–3	Nivolumab + oxaliplatin-based chemotherapy	1st-line, HER2–unresectable/recurrent GC; improved PFS; OS benefit not significant; favorable long-term outcomes.
CheckMate-649	3	Nivolumab + chemotherapy vs. chemo alone	1st-line advanced GC/GEJ/esophageal adenocarcinoma; improved OS and PFS, especially in PD-L1 CPS ≥ 5; established new standard of care.
CheckMate-032	2	Nivolumab ± ipilimumab	Heavily pretreated advanced GC/GEJ cancer; ORR up to 24% (NIVO1+IPI3); durable responses; comparable OS across arms.

**Table 2 antibodies-14-00058-t002:** Summary of trials evaluating immunotherapy in HCC.

Drug class	Name of Study	Phase	Article	Drug
ICI	GO30140 study	Phase Ib	Lee et al. [34]	Atezolizumab + bevacizumab (vs. sorafenib)
ICI	--	Phase II	Sangro B, Gomez et al. [28]	Tremelimumab
ICI	CheckMate 040	Phase II	El-Khoueiry et al.[29]	Nivolumab
ICI	Keynote 224	Phase II	Zhu et al. [32]	Pembrolizumab
ICI	Himalaya trial	Phase III	Abou-Alfa et al. [38]	Tremelimumab + durvalumab
ICI	CheckMate 459	Phase III	Yau et al. [30]	Nivolumab
ICI	Keynote 240	Phase III	Finn, Ryoo et al. [41]	Pembrolizumab
ICI	IMbrave050	Phase III	Qin et al. [36]	Atezolizumab + bevacizumab (vs active surveillance)
ICI	IMbrave150	Phase III	Finn et al. [35]	Atezolizumab + bevacizumab (vs sorafenib)
ICI	Keynote-937	Phase III	NCT03867084	Pembrolizumab
ICI	EMERALD-2	Phase III	NCT03847428	Durvalumab +/- bevacizumab
ICI + TKI		Phase Ib	Bang et al. [39]	Durvalumab + remucirumab
ICI + TKI		Phase I/II	NCT04212221	MGD013 + brivanib

**Table 3 antibodies-14-00058-t003:** Key clinical trials shaping CRC immunotherapy.

Trial Name	Combination Therapy	Target Mechanism
** *BREAKWATER* **	Encorafenib + Cetuximab ± Chemotherapy	BRAF V600E-targeted therapy
** *FRESCO-2* **	Fruquintinib + Standard Care	Anti-angiogenesis
***COMMIT*** [51]	Atezolizumab + Bevacizumab ± Chemotherapy	PD-L1 + VEGF blockade
***RENMIN-215*** [52]	Fecal Microbiota Transplantation + ICIs	Gut microbiome modulation
***Zabadinostat + Nivolumab*** [53]	HDAC Inhibitor + PD-1 Blockade	Epigenetic modulation

**Table 4 antibodies-14-00058-t004:** Key clinical trials in pancreatic cancer immunotherapy.

Trial Name	Phase	Therapy Evaluated	Patient Population/Key Findings
** *KEYNOTE-158* **	II	Pembrolizumab monotherapy	MSI-H/dMMR advanced PDAC; ORR 18.2%, median PFS 2.0 months, and median OS 4.0 months
** *Chemo4METPANC* **	II	Cemiplimab + Motixafortide + Gemcitabine/Nab-Paclitaxel	Metastatic PDAC; in a pilot study with 11 patients: 7 partial responses, 3 stable disease, 1 progression; promising early efficacy
** *Autogene Cevumeran* **	I	Personalized mRNA vaccine + Atezolizumab	Resected PDAC; among 16 patients, 8 responders showed no recurrence at 18 months; non-responders had a median recurrence-free survival of 13.4 months
** *IMM-101 Study* **	II	IMM-101 + Gemcitabine vs. Gemcitabine alone	Advanced PDAC; combination therapy was well-tolerated; suggested potential survival benefit over gemcitabine alone

Note: MSI-H/dMMR refers to microsatellite instability-high/mismatch repair-deficient tumors, which are present in a small subset of PDAC patients.

## Data Availability

No new data were created or analyzed in this study. Data sharing is not applicable to this article.

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
