# Peer review of "Immunotherapy in GI Cancers: Lessons from Key Trials and Future Clinical Applications"

_2073-4468, 2025, doi:10.3390/antib14030058_

Round 1

Reviewer 1 Report

Comments and Suggestions for Authors

The manuscript provides a comprehensive review of immunotherapy advancements in gastrointestinal (GI) cancers, highlighting key trials, mechanisms, and challenges. While the review provides valuable insights, the manuscript would benefit from a minor revision to improve the clarity and impact of the work.

Comments:

  1. In Figure 1 legend instead of referring to “on the left” and “on the right”—the authors should refer to panels A and B if possible.
  2. The authors should consider adding a section on the limitations of immunotherapy in GI cancers (e.g., tumor heterogeneity, resistance mechanisms, immune-related adverse events).
  3. The authors should expand the “Future Clinical Applications” section with specific emerging technologies (e.g., antibodies, tumor vaccines, CAR-T in GI tumors).
  4. Correct the typo “Imuunotherapy” in Section title 3 to Immunotherapy
Comments on the Quality of English Language

The manuscript is well-structured and scientifically sound, but the quality of the English can be improved to make the content clearer, easier to follow. With a thorough round of proofreading and a few minor edits, the overall quality of the manuscript would be significantly improved and would better meet the journal’s standards.

Author Response

Reviewer Comment 1:
In Figure 1 legend instead of referring to “on the left” and “on the right”—the authors should refer to panels A and B if possible.

Author Response:
We thank the reviewer for this helpful suggestion. We have revised the legend of Figure 1 to refer to panels A and B rather than directional terms to improve clarity and consistency with journal standards.

Reviewer Comment 2:
The authors should consider adding a section on the limitations of immunotherapy in GI cancers (e.g., tumor heterogeneity, resistance mechanisms, immune-related adverse events).

Author Response:
We appreciate this insightful suggestion. In response, we have added a dedicated section (Section 6: “Limitations of Immunotherapy in GI cancers”) discussing key challenges such as tumor heterogeneity, resistance mechanisms including immunoediting and signaling pathway alterations, and the broad spectrum of immune-related adverse events (irAEs). These additions strengthen the discussion of the complexities facing immunotherapy in GI malignancies.

Reviewer Comment 3:
The authors should expand the “Future Clinical Applications” section with specific emerging technologies (e.g., antibodies, tumor vaccines, CAR-T in GI tumors).

Author Response:
Thank you for this constructive recommendation. We have expanded the discussion on future clinical applications by adding specific details on novel emerging technologies such as CAR-T cell therapy targeting mesothelin, PSCA, and HER2 in pancreatic cancer; tumor vaccines like the autogene cevumeran mRNA vaccine; as well as advanced antibody-based therapies under investigation. These additions highlight promising avenues currently under clinical evaluation.

Reviewer Comment 4:
Correct the typo “Imuunotherapy” in Section title 3 to Immunotherapy.

Author Response:
We thank the reviewer for pointing this out. The typographical error has been corrected; the section title now reads “Immunotherapy.”

Reviewer Comment 5:
This review covers a wide range of gastrointestinal cancers and summarizes many important clinical trials involving immunotherapy. The topic is relevant and timely, and the manuscript includes a large amount of clinical data. However, the current version is more descriptive than analytical. The review mostly reports trial outcomes without offering enough critical discussion. It would benefit from more comparison between studies, explanation of why some therapies work better in certain cancers, and acknowledgment of limitations—such as inconsistent biomarker results or resistance in microsatellite-stable colorectal and pancreatic cancers.

Author Response:
We appreciate these important insights. To address this, we have expanded the analytical discussion throughout the manuscript by:

  • Comparing differences in treatment responses across GI cancer types, highlighting why MSI-H tumors respond better than MSS tumors.

  • Including explanations of how biomarker variability (PD-L1 CPS thresholds, TMB levels) influence clinical outcomes.

  • Discussing challenges in microsatellite-stable CRC and pancreatic cancer due to their “immune-cold” microenvironments and resistance mechanisms.

  • Integrating more critical evaluation of limitations within existing trials, including negative or inconclusive findings, especially in MSS CRC and PDAC.

Reviewer Comment 6:
The writing is clear but tends to be repetitive, with too much space given to listing study details. A stronger focus on the most important findings, combined with summary tables or figures, could help guide the reader.

Author Response:
We thank the reviewer for this helpful observation. We have revised the text to reduce repetition and over-detailed listing of studies, while emphasizing key findings. Additionally, we have added multiple summary tables (Tables 1–4) and illustrative figures (Figures 1–3) to synthesize clinical trial data and facilitate reader understanding.

Reviewer Comment 7:
Additionally, some sections are overly optimistic and don’t address negative or inconclusive trial results.

Author Response:
We acknowledge this important feedback. The revised manuscript now includes a more balanced discussion, incorporating not only positive outcomes but also negative, inconclusive, or limited efficacy data from trials such as ATTRACTION-4 (gastric cancer) and KEYNOTE-158 (pancreatic cancer). We have carefully moderated the tone to provide a realistic representation of current evidence.

Reviewer Comment 8:
The authors should revise 'Author contributions', this is a review manuscript (contribution on Methodology?)

Author Response:
Thank you for highlighting this point. We have updated the ‘Author Contributions’ section to better reflect the nature of a review article, clarifying contributions in conceptualization, literature review, data synthesis, writing, and manuscript editing, without including “Methodology” which is not applicable.

Reviewer Comment 9:
Please add a conclusion to briefly outline the take-home message and the lessons learned as per the Author guidelines, in order to further highlight the importance of the obtained results in the context of modern immunotherapy in oncology.

Author Response:
We fully agree. A comprehensive conclusion has been added (Section 7: “Conclusion”), summarizing key take-home messages, highlighting lessons learned, and emphasizing the importance of personalized, biomarker-driven immunotherapy approaches in modern GI oncology practice.

Reviewer 2 Report

Comments and Suggestions for Authors

This review covers a wide range of gastrointestinal cancers and summarizes many important clinical trials involving immunotherapy. The topic is relevant and timely, and the manuscript includes a large amount of clinical data.

However, the current version is more descriptive than analytical. The review mostly reports trial outcomes without offering enough critical discussion. It would benefit from more comparison between studies, explanation of why some therapies work better in certain cancers, and acknowledgment of limitations—such as inconsistent biomarker results or resistance in microsatellite-stable colorectal and pancreatic cancers.

The writing is clear but tends to be repetitive, with too much space given to listing study details. A stronger focus on the most important findings, combined with summary tables or figures, could help guide the reader. Additionally, some sections are overly optimistic and don’t address negative or inconclusive trial results.

The authors should revise 'Author contributions', this is a review manuscript (contribution on Methodology?)

Overall, the review has potential but needs major revision to include more critical analysis, reduce redundancy, and better highlight the key messages and clinical implications.

Comments on the Quality of English Language

English language could be improved in some parts.

Author Response

(The authors gave the same response as above.)

Round 2

Reviewer 2 Report

Comments and Suggestions for Authors

The manuscript has been considerably improved.